

# Phospholipase A$_2$ from krait *Bungarus fasciatus* venom induces human cancer cell death in vitro

Thien V. Tran[1,2], Andrei E. Siniavin[3], Anh N. Hoang[2,4], My T.T. Le[5], Chuong D. Pham[5], Trung V. Phung[6], Khoa C. Nguyen[2,4], Rustam H. Ziganshin[7], Victor I. Tsetlin[8], Ching-Feng Weng[9] and Yuri N. Utkin[3]

[1] Tra Vinh University, Tra Vinh City, Vietnam
[2] Graduate University of Science and Technology VAST, Hanoi, Vietnam
[3] Laboratory of Molecular Toxinology, Shemyakin-Ovchinnikov Institute of Bioorganic Chemistry RAS, Moscow, Russian Federation
[4] Institute of Applied Materials Science VAST, Ho Chi Minh City, Vietnam
[5] Faculty of Applied Sciences, Ton Duc Thang University, Ho Chi Minh City, Vietnam
[6] Center for Research and Technology Transfer VAST, Ho Chi Minh City, Vietnam
[7] Shemyakin-Ovchinnikov Institute of Bioorganic Chemistry RAS, Moscow, Russian Federation
[8] Department of Molecular Neuroimmune Signalling, Shemyakin-Ovchinnikov Institute of Bioorganic Chemistry RAS, Moscow, Russian Federation
[9] Department of Life Science and Institute of Biotechnology, National Dong Hwa University, Shoufeng, Hualien, Taiwan

Corresponding author
Yuri N. Utkin, utkin@mx.ibch.ru

## ABSTRACT

**Background:** Snake venoms are the complex mixtures of different compounds manifesting a wide array of biological activities. The venoms of kraits (genus Bungarus, family Elapidae) induce mainly neurological symptoms; however, these venoms show a cytotoxicity against cancer cells as well. This study was conducted to identify in *Bungarus fasciatus* venom an active compound(s) exerting cytotoxic effects toward MCF7 human breast cancer cells and A549 human lung cancer cells.

**Methods:** The crude venom of *B. fasciatus* was separated by gel-filtration on Superdex HR 75 column and reversed phase HPLC on C18 column. The fractions obtained were screened for cytotoxic effect against MCF7, A549, and HK2 cell lines using colorimetric assay with the tetrazolium dye MTT- 3-(4,5-dimethylthiazol-2-yl)-2,5-diphenyltetrazolium bromide. The primary structure of active protein was established by ultra high resolution LC-MS/MS. The molecular mechanism of the isolated protein action on MCF7 cells was elucidated by flow cytometry.

**Results:** MTT cell viability assays of cancer cells incubated with fractions isolated from *B. fasciatus* venom revealed a protein with molecular mass of about 13 kDa possessing significant cytotoxicity. This protein manifested the dose and time dependent cytotoxicity for MCF7 and A549 cell lines while showed no toxic effect on human normal kidney HK2 cells. In MCF7, flow cytometry analysis revealed a decrease in the proportion of Ki-67 positive cells. As Ki-67 protein is a cellular marker for proliferation, its decline indicates the reduction in the proliferation of MCF7 cells treated with the protein. Flow cytometry analysis of MCF7 cells stained with propidium iodide and Annexin V conjugated with allophycocyanin showed that a probable mechanism of cell death is apoptosis. Mass spectrometric studies

showed that the cytotoxic protein was phospholipase A$_2$. The amino acid sequence of this enzyme earlier was deduced from cloned cDNA, and in this work it was isolated from the venom as a protein for the first time. It is also the first krait phospholipase A$_2$ manifesting the cytotoxicity for cancer cells.

## INTRODUCTION

Cancer is the second leading cause of death in the world. Despite advances in the development of new drugs, the search for new effective medicines remains a challenging task. The main problem is a high toxicity of the existing drugs to the normal cells. Snake venoms contain many bioactive proteins manifesting diverse biological activities. Some of them were shown to possess cytotoxic activity against tumor cells. Anti-cancer activity was demonstrated for venoms of snakes from different genera and species (*Li, Huang & Lin, 2018*). There are numerous communications on cytotoxic effects of venoms from cobras, vipers and pit-vipers on tumor cells (*Zainal Abidin et al., 2019*; *Nalbantsoy et al., 2017*; *Ghazaryan et al., 2015*). Several proteins, including cytotoxins (*Dubovskii & Utkin, 2015*), L-amino acid oxidases (*Salama et al., 2018*), phospholipases A$_2$ (*Sobrinho et al., 2016*), disintegrins (*Arruda Macêdo, Fox & De Souza Castro, 2015*) and others manifesting anti-proliferative activity were isolated from these venoms. These proteins themselves, due to their inherent adverse properties (high molecular mass, high toxicity, etc.), can hardly be used as medicines. However, their active fragments may well be used for this purpose. The adverse effects of the venom proteins can be greatly diminished by use of the targeted drug delivery systems. For example, liposomal delivery of disintegrins substantially increased their therapeutic potential (*David et al., 2018*).

The anti-cancer activity of kraits' venoms is not so well studied as that of other venoms. Thus, it was found that β-bungarotoxin from krait *Bungarus multicinctus* venom showed the concentration- and time-dependent cytotoxicity against human neuroblastoma SK-N-SH cells (*Cheng, Wang & Chang, 2008*). Moreover, the cytotoxic effect was localized on B-subunit of β-bungarotoxin. L-Amino acid oxidases isolated from *B. fasciatus* (*Wei et al., 2009*) and *B. multicinctus* (*Lu et al., 2018*) venoms manifested strong cytotoxicity against different cancer cell lines. A protease inhibitor like protein-1 (PILP-1) from *B. multicinctus* venom was found to induce apoptotic death of human leukemia U937 cells (*Liu & Chang, 2010*). The more detailed studies showed that PILP-1-induced down-regulation of a disintegrin and metalloprotease 17 (ADAM17) which resulted in inactivation of Lyn/Akt pathways. The mitochondrion-mediated apoptosis of U937 cells was thus activated. From *B. fasciatus* krait venom, a protein BF-CT1 possessing capacity to induce Ehrlich ascites carcinoma (EAC) and U937 leukemic cell death was isolated (*Bhattacharya et al., 2013*). BF-CT1 had molecular mass of 13 kDa and induced apoptosis in EAC in vivo and in U937 cell line in vitro.

The above studies indicated that krait venoms have some anti-cancer potential. In this work we present the data on activity-guided isolation from Vietnamese krait *B. fasciatus* venom and characterization of a phospholipase $A_2$ manifesting cytotoxic activity against human MCF7 and A549 cell lines.

## MATERIALS AND METHODS

### Materials

#### Snake venom

Crude krait *B. fasciatus* venom (Vinh Son, Vinh Tuong, Vinh Phuc Province, Vietnam) was obtained as previously described (*Ziganshin et al., 2015*). The venom was collected from several tens of snake specimens at the farm owned by professional snake breeder Mr. Ha Van Tien by farm team members. It was lyophilized and stored at $-20\,^\circ$C until use.

#### Cell lines

The human breast cancer cell line MCF7 (Catalog number: HTB-22), the human breast cancer cells BT-474 (Catalog number: HTB-20), the human breast cancer cells SK-BR-3 (Catalog number: HTB-30), the human prostate cancer cells PC-3 (Catalog number: CRL-1435), the human prostate cancer cells LNCaP (Catalog number: CRL-1740), the human lung cancer cells A549 (Catalog number: CCL-185EMT), and renal tubular epithelial HK-2 cells (a proximal tubular cell line derived from normal kidney; Catalog number: CRL-2190) were purchased from the American Type Culture Collection (ATCC, Rockville, MD, USA) and stored in liquid nitrogen until use.

### Methods

#### Fractionation of krait B. fasciatus venom

Crude krait *B. fasciatus* venom was separated by gel-filtration on the Superdex® 75 10/300 GL column (1 × 30 cm; GE Healthcare Bio-Sciences, Pittsburgh, PA, USA) equilibrated with the 0.1 M ammonium acetate buffer (pH 6.2). The column was eluted at flow rate of 0.5 ml/min and the eluate was monitored by measuring optical density at 226 nm (Fig. 1A). The fractions obtained were freeze-dried and used for activity measurements.

Fraction 3 (Fig. 1A) was further separated by reversed-phase high-performance liquid chromatography on the Jupiter C18 column (10 × 250 mm) in a gradient of 25–40% acetonitrile in 75 min in the presence of 0.1% trifluoroacetic acid, at a flow rate of 2.0 ml/min. The eluate was monitored by measuring optical density at 275 nm (Fig. 1B). The obtained fractions were freeze-dried and used for activity measurements.

#### Cell culture

Both MCF7 and A549 cells were cultured in DMEM supplemented with 10% FBS, 1% antibiotics (100 U/mL of penicillin and 100 μg/mL of streptomycin). HK2 cells were grown in DMEM/F12 supplemented with 10% FBS, 1% antibiotics (100 U/mL of penicillin and 100 μg/mL of streptomycin), and 5 ng/mL of human recombinant epidermal growth factor (EGF, Gibco, Waltham, MA, USA). All of cell lines were kept at 37 $^\circ$C in a humidified atmosphere of 5% $CO_2$ incubator.

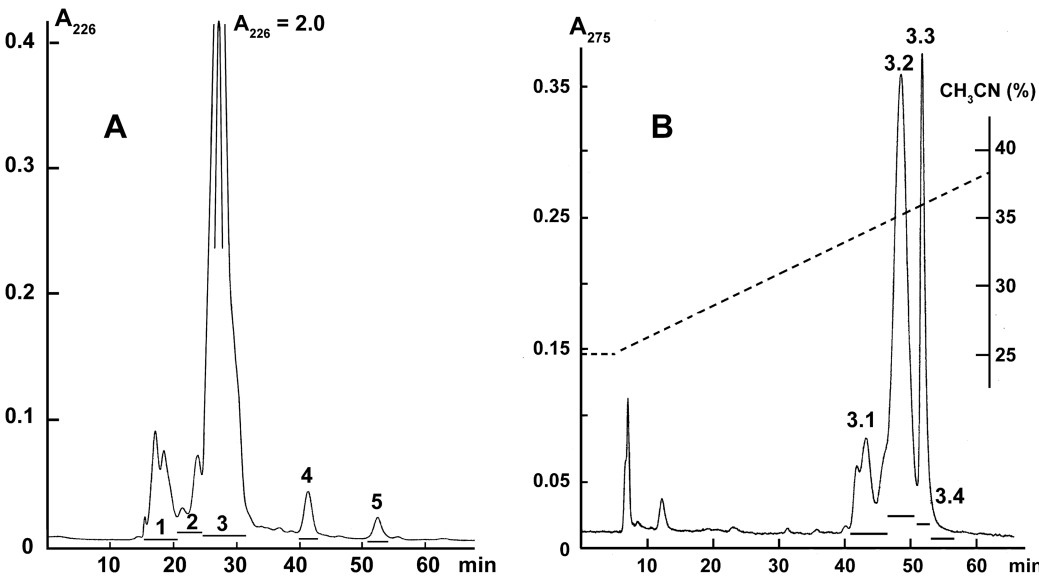

**Figure 1 Fractionation of *B. fasciatus* venom.** (A) Gel filtration of crude venom on the Superdex® 75 10/300 GL column (1 × 30 cm) equilibrated with the 0.1 M ammonium acetate buffer (pH 6.2). Flow rate 0.5 ml/min. The eluate was monitored by spectrophotometry (OD = 226 nm). Horizontal bars indicate the collected fractions. (B) Reversed phase chromatography of Fraction 3 (From A) on the Jupiter C18 column (10 × 250 mm) in a gradient of 25–40% acetonitrile in 75 min in the presence of 0.1% tri-fluoroacetic acid, at a flow rate of 2.0 ml/min. The eluate was monitored by spectrophotometry (OD = 275 nm). Horizontal bars indicate the collected fractions.

## Determination of cytotoxicity by MTT assay

Cell viability was examined using the colorimetric MTT [3-(4, 5-dimethyl-2-thiazolyl)-2, 5-diphenyl-2H-teyrazolium bromide] (Sigma–Aldrich, St. Louis, MO, USA) assay, as previously described (*Mosmann, 1983*). Briefly, the cell lines were seeded at $1 \times 10^4$ cells per well in 96-well plates for 24 h at 37 °C in a humidified atmosphere of 5% $CO_2$ to allow cell attachment. Cells were treated with 1, 10, 50, and 100 μg/mL of crude venom and various fractions or 1, 5, and 10 μg/mL of cisplatin (Sigma–Aldrich, St. Louis, MO, USA) for 24, 48 or 72 h, and then 10 μL/well of MTT reagent was added into the wells and further incubated for additional 4 h. The supernatant was decanted, and dimethyl sulfoxide (DMSO, 100 μL/well) was added to allow formazan solubilization. The optical density value was measured at 570 nm using a SpectraMax-190 96-well plate reader (Molecular Devices, Sunnyvale, CA, USA). The percentage of viable cells was determined from a comparison with untreated control.

The significance of differences between experimental and control groups was analyzed by *t*-test: Two-Sample Assuming Equal Variances using Microsoft Excel 2016 MSO (v.1902; Microsoft Corporation, Redmond, WC, USA) program. All results are presented as the mean ± SEM (standard error of the mean).

## Morphological studies by phase contrast microscopy

Cells of both lines were seeded and treated with the venom and fractions as described in the previous section. Changes in the cell morphology were observed using phase contrast

microscopy with Zeiss Axio Vert 25C (Carl Zeiss Microscopy GmbH, Göttingen, Germany).

### Flow cytometry (FACS analysis)

#### Cell proliferation analysis

To assess cell proliferation, flow cytometric analysis was performed using phycoerythrin-conjugated monoclonal anti-Ki-67 antibody (clone Ki-67; Sony Biotechnology, San Jose, CA, USA). For intracellular staining, cells were collected and washed twice with ice-cold phosphate-buffered saline, re-suspended in cold 70% ethanol and incubated at −20 °C for 1 h. Then the cells were washed twice with phosphate-buffered saline containing 0.3% bovine serum albumin, incubated with anti-Ki-67 antibody at room temperature for 30 min and analyzed using BD FACSCalibur flow cytometer (BD Biosciences, San Jose, CA, USA).

#### Apoptosis analysis

MCF7 cells were seeded at a density $2,5 \times 10^5$ per well of a 12-well plate and incubated overnight for cell attachment. Next, MCF7 cells were treated with phospholiapse $A_2$ (fraction 3.3) at various concentration for 24 h. After the treatment, cells were washed twice with phosphate-buffered saline, re-suspended in stain solution containing allophycocyanin (APC)-conjugated Annexin V and propidium iodide (PI) (both from Biolegend, San Diego, CA, USA), incubated for 15 min at room temperature, and analyzed using BD FACSCalibur flow cytometer equipped with 488- and 640-nm lasers. The data were analyzed using FlowJo 10 software (FlowJo LLC, Ashland, OR, USA).

### Mass-spectrometry measurements

#### MALDI mass-spectrometric analysis

MALDI-TOF mass spectrometry analyses were performed using Ultraflex TOF/TOF mass spectrometer (Bruker Daltonik GmbH, Bremen, Germany). Mass spectra were recorded in linear mode of positively charged ions in the m/z range of 5–20 kDa using 2,5-dihydroxybenzoic acid (20 mg/ml, acetonitrile/0.1% TFA 1:1, v/v) as a matrix. The mass spectrometry data were processed using Bruker Daltonics Flex Analysis 2.4 software.

#### Reduction, alkylation and digestion of the protein

Reduction, alkylation and digestion of the protein were performed as previously described (*Kulak et al., 2014*) with minor modifications. Briefly, sodium deoxycholate (SDC) reduction and alkylation buffer pH 8.5 was added to a protein sample (10 μg) so that the final concentration of protein, TRIS, SDC, TCEP and 2-chloroacetamide were 0.5 mg/ml, 100 mM, 1% (w/v), 10 mM and 40 mM, respectively. The solution was heated for 10 min at 95 °C, cooled down to a room temperature and the equal volume of trypsin solution in 100 mM TRIS pH 8.5 was added in a 1:100 (w/w) ratio. Digestion was carried out overnight at 37 °C.

#### Tryptic peptides desalting

Desalting of peptides was carried out using SDB-RPS StageTips that were prepared as described earlier (*Rappsilber, Mann & Ishihama, 2007*). Briefly, two pieces of 3M Empore

SDB-RPS membrane were stamped out using blunt-ended Hamilton needle (part# 91014: Metal (N) Hub, Point Style 3, gauge 14) and forced into the 200 μl pipette tip end by a piece of 1/16″ OD PEEK tubing (1535; Upchurch Scientific). A two mL microcentrifuge tube with an opening punctured in the tube's lid (O-tube) was used as a StageTip holder—SDB-RPS StageTip and O-tube comprises the Spin-unit. After overnight digestion, peptide solution was acidified by equal volume of 2% (v/v) TFA and peptides were loaded on StageTip by centrifugation at 200 g. StageTip was washed by 50 μl ethylacetate/50 μl 0.2% (v/v) TFA three times. Peptides were eluted by 60 μl 80% (v/v) acetonitrile, 5% (v/v) $NH_4OH$, lyophilized and stored at −80 °C. Before analyses peptides were dissolved in 20 μl of 2% (v/v) acetonitrile, 0.1% (v/v) TFA and sonicated for 2 min.

*LC-MS analyses*

Peptides and proteins were separated on a 50 cm 75 μm inner diameter column packed in-house with Aeris Peptide XB-C18 2.6 μm resin (Phenomenex, Torrance, CA, USA). Reverse-phase chromatography was performed with an Ultimate 3000 Nano LC System (Thermo Fisher Scientific, Waltham, MA, USA), which was coupled to a Q Exactive HF benchtop Orbitrap mass spectrometer (Thermo Fisher Scientific, Waltham, MA, USA) via a nanoelectrospray source (Thermo Fisher Scientific, Waltham, MA, USA). The mobile phases were: (A) 0.1% (v/v) formic acid in $H_2O$ and (B) 0.1% (v/v) formic acid, 80% (v/v) acetonitrile, 19.9% (v/v) $H_2O$. Samples were loaded onto a trapping column (100 μm internal diameter, 20 mm length, packed in-house with Aeris Peptide XB-C18 2.6 μm resin (Phenomenex, Torrance, CA, USA)) in mobile phase A at flow rate 5 μl/min for 5 min and eluted with a linear gradient of mobile phase B (5–45% B in 60 min—for peptides, and 15–60% B in 24 min—for proteins) at a flow rate of 350 nl/min. Column temperature was kept at 40 °C. Peptides were analyzed on the Q Exactive HF benchtop Orbitrap mass spectrometer (Thermo Fisher Scientific, Waltham, MA, USA), with one full scan (300–1,400 m/z, $R$ = 60,000 at 200 m/z) at a target of 3e6 ions, followed by up to 15 data-dependent MS/MS scans with higher-energy collisional dissociation (HCD) (target 1e5 ions, max ion fill time 60 ms, isolation window 1.2 m/z, normalized collision energy (NCE) 28%, underfill ratio 2%), detected in the Orbitrap ($R$ = 15,000 at fixed first mass 100 m/z). Other settings: charge exclusion—unassigned, 1, >6; peptide match—preferred; exclude isotopes—on; dynamic exclusion—30 s was enabled. Proteins were analyzed on the same mass spectrometer with the following parameters: In-source CID = 0.0 eV; scan range—350–2000 m/z; $R$ = 120,000 at 200 m/z; AGC target of 3e6; max ion fill time 100 ms.

*Data analyses*

MS raw files were analyzed by PEAKS Studio 8.5 (Bioinformatics Solutions Inc., Waterloo, ON, Canada) (*Ma et al., 2003*) and peak lists were searched against Serpentes Uniprot-Tremble FASTA (canonical and isoform) database version of May 2018 (144,954 entries) with cysteine carbamidomethylation as a fixed modification and methionine oxidation and asparagine and glutamine deamidation as variable modifications. False discovery rate was set to 0.01 for peptide-spectrum matches and was

determined by searching a reverse database. No enzyme specificity was set in the database search. Peptide identification was performed with an allowed initial precursor mass deviation up to 10 ppm and an allowed fragment mass deviation 0.05 Da.

# RESULTS

## Cytotoxicity studies

We have recently shown that krait *B. fasciatus* venom from Vinh Phuc province (Vietnam) possessed the capacity to inhibit proliferation of the human breast cancer cell line MCF7 and the human lung cancer cell A549 (*Tran et al., 2019*). To isolate an active compound, the venom was subjected to fractionation by means of liquid chromatography. The gel-filtration on Superdex 75 column was used as a first step. As a result, five fractions were obtained (Fig. 1A). After freeze-drying, the fractions were screened for cytotoxicity against MCF7 and A549 cell lines (Fig. 2). It was found that only fraction 3 was able to manifest the cytotoxicity for both cell lines. The cytotoxic effect of fraction 3 was time- and dose-dependent. After 72 h of incubation at a concentration of 100 μg/mL, the percentage of living MCF7 cells decreased significantly to less than 20% of the control and that of living A549 cells—to 16% of the control. Analysis of the active fraction by MALDI mass spectrometry revealed the presence of several proteins in this fraction (Fig. 3). Very intensive signals were observed at m/z around 13,000 which correspond apparently to single charged ions. In *B. fasciatus* venom only phospholipases $A_2$ may possess molecular mass in this range. The signals at m/z around 6,500 are double charged ions of the same proteins, while those in m/z range 7,300–7,400 may correspond to three finger toxins. The active fraction 3 was further separated by reversed-phase chromatography on C18 column, and as a result, four fractions were obtained (Fig. 1B). Three main fractions obtained were analyzed by MALDI mass spectrometry (Fig. 3). All fractions analyzed displayed single charged signal around m/z 13,000 and double-charged signals around m/z 6500. These data indicate that fractions 3.2 and 3.3 contain proteins with molecular masses of 13,018 and 13,093 Da, respectively. After freeze-drying the HPLC fractions were assayed for the cytotoxicity using MTT test. Only fraction 3.3 manifested the cytotoxicity for both tested cell lines (Fig. 2). The yield of the active protein was about 10% of the crude venom. The anticancer effect of this fraction increased with increasing concentration and incubation time. After 72 h of incubation at a concentration of 100 μg/mL of fraction 3.3, the percentage of living MCF7 cells decreased to 56.7% of control and that of living A549 cells—to 57%. All other fractions could not induce the cancer cell death even after incubation for 72 h at concentration of 100 μg/mL. We have tested the effect of fraction 3.3 on several other human cancer cells: the human breast cancer cells BT-474 and SK-BR-3 as well as the human prostate cancer cells PC-3 and LNCaP. Fraction 3.3 at concentration of 100 μg/mL reduced the viability of BT-474, PC-3 and LNCaP cells by 39, 20 and 12%, respectively, while produced no effect on SK-BR-3 cells. Interestingly, neither crude venom nor any fraction including fractions 3 and 3.3 produced a noticeable effect on the noncancerous human kidney epithelial HK2 cells at 24, 48 or 72 h of treatment (Fig. 4). In human chemotherapy, cisplatin is among the most active antitumor agent which are used to treat breast, lung, ovarian, and bladder cancers (*Kelland, 2007*). As expected, cisplatin as

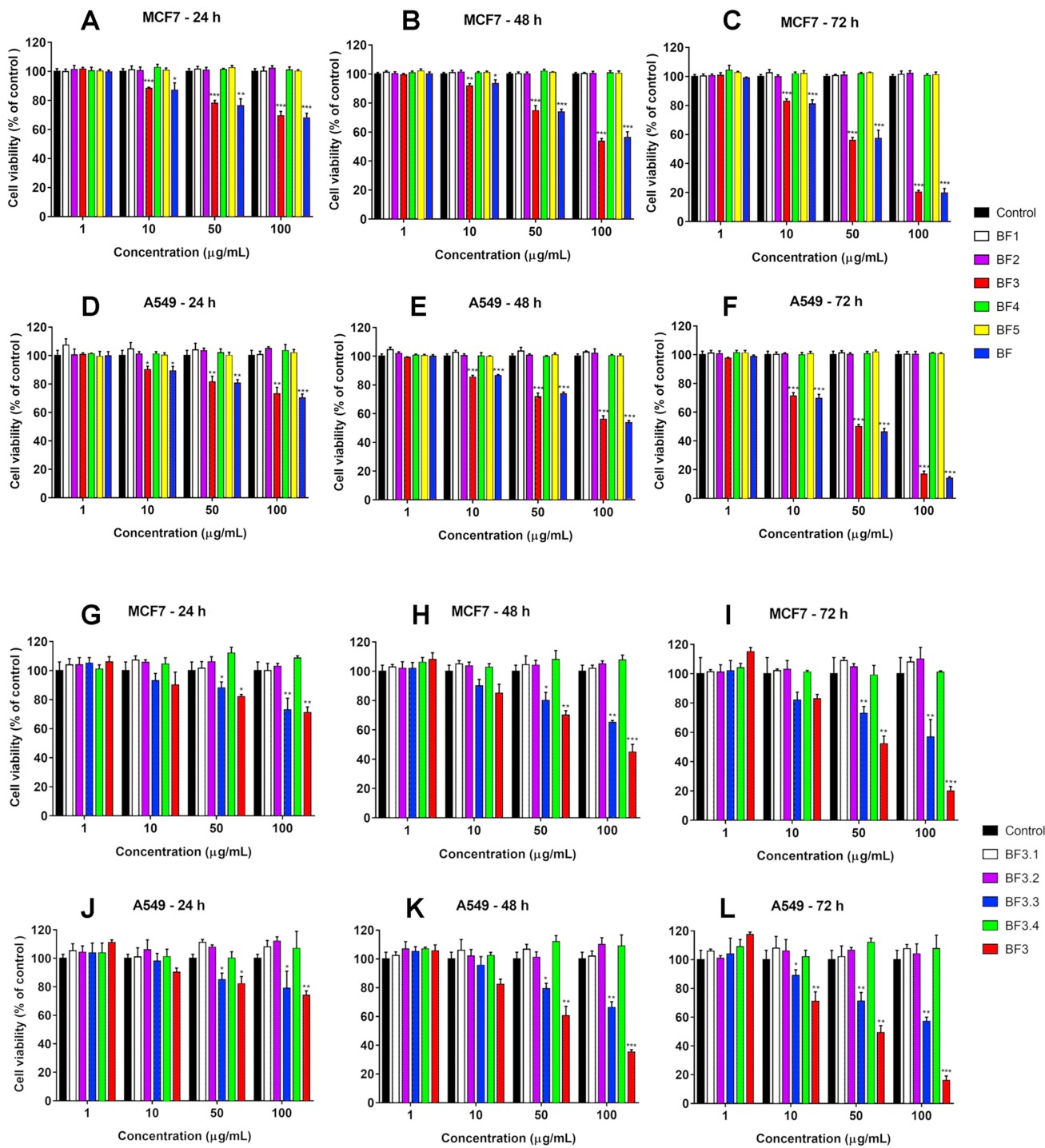

**Figure 2 Cytotoxicity of fractions obtained from *B. fasciatus* venom by gel-filtration and reversed phase chromatography against human MCF7 and A549 cell lines.** Cell viability was examined using the colorimetric MTT assay. The percentage of viable cells was determined from a comparison with untreated control. All results are presented as the mean ± SEM (standard error of the mean). $n = 3$. The significance of differences between experimental and control groups was analyzed by *t*-test: Two-Sample Assuming Equal Variances using the Microsoft Excel 2016 MSO program. The complete statistical data are reported in Supplemental Materials 1 and 2. Here, $p < 0.05$, $p < 0.01$ and $p < 0.001$ are indicated by *, ** and ***, respectively. (A–F) Gel-filtration fractions. BF1–BF5 correspond to fractions 1–5 from Fig. 1A. BF—*B. fasciatus* venom. (G–L) Fractions obtained after reversed phase chromatography (Fig. 1B). BF3.1–BF3.4 correspond to fractions 3.1–3.4 from Fig. 1B. BF3 fraction 3 from Fig. 1A.

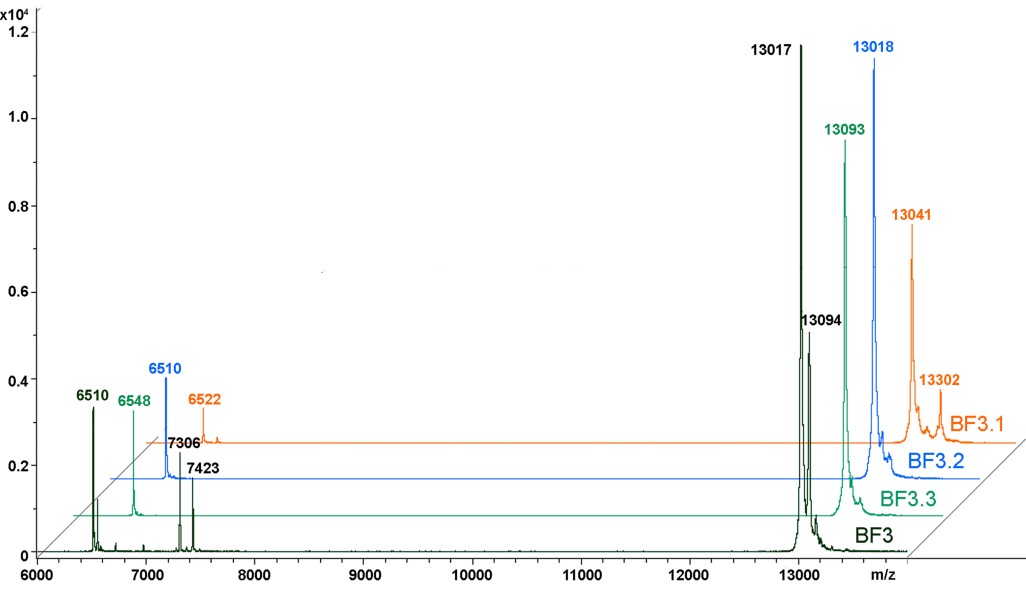

**Figure 3 MALDI mass spectroscopy analysis of *B. fasciatus* venom fractions.** MALDI-TOF mass spectrometry measurements were performed using Ultraflex TOF/TOF mass spectrometer. The mass spectrometry data were processed using Bruker Daltonics Flex Analysis 2.4 software. BF3—fraction 3 obtained after gel-filtration of crude venom on Superdex® 75 10/300 GL column (Fig. 1A). BF3.1, BF3.2 and BF3.3—fractions 3.1, 3.2 and 3.3, respectively obtained by reversed phase chromatography of gel-filtration fraction 3 on Jupiter C18 column (Fig. 1B).

the positive control induced cell cytotoxicity in all cell lines including human normal kidney HK2 cells (Fig. 5). These results indicate that krait venom and isolated compound are toxic to cancer cells while produce practically no effect on normal cells.

## Morphological studies using phase contrast microscopy

In this work, the morphological alterations of the human breast cancer cells MCF7 and the human lung cancer cells A549 treated with crude *B. fasciatus* venom and fractions were identified under a phase contrast microscope. In the control group, untreated cells spread regularly on the bottom of the culture plates and grew to near confluence (Figs. 6A and 6D). Untreated cells were uniform in size, appeared elongated, and attached smoothly on the plate surface.

After exposure to the venom and fractions, cells showed drastic changes in their overall morphology. Following treatment with the venom or fractions, cell shrinkage, loss of cell adhesion and reduced cell density were clearly observed (Fig. 6). The cell shape changed from elongated to round with increased intercellular spaces. The cells detached from the culture plates and floated in the medium. Similar effects are characteristic for cytotoxic compounds.

## Mass spectrometry studies

As discussed above some fractions were analyzed by MALDI mass spectrometry. It was found that cytotoxic fraction 3.3 contains protein with molecular mass of 13,093 Da (Fig. 3). In the spectrum of this fraction as well as in the spectra of other fractions, on the

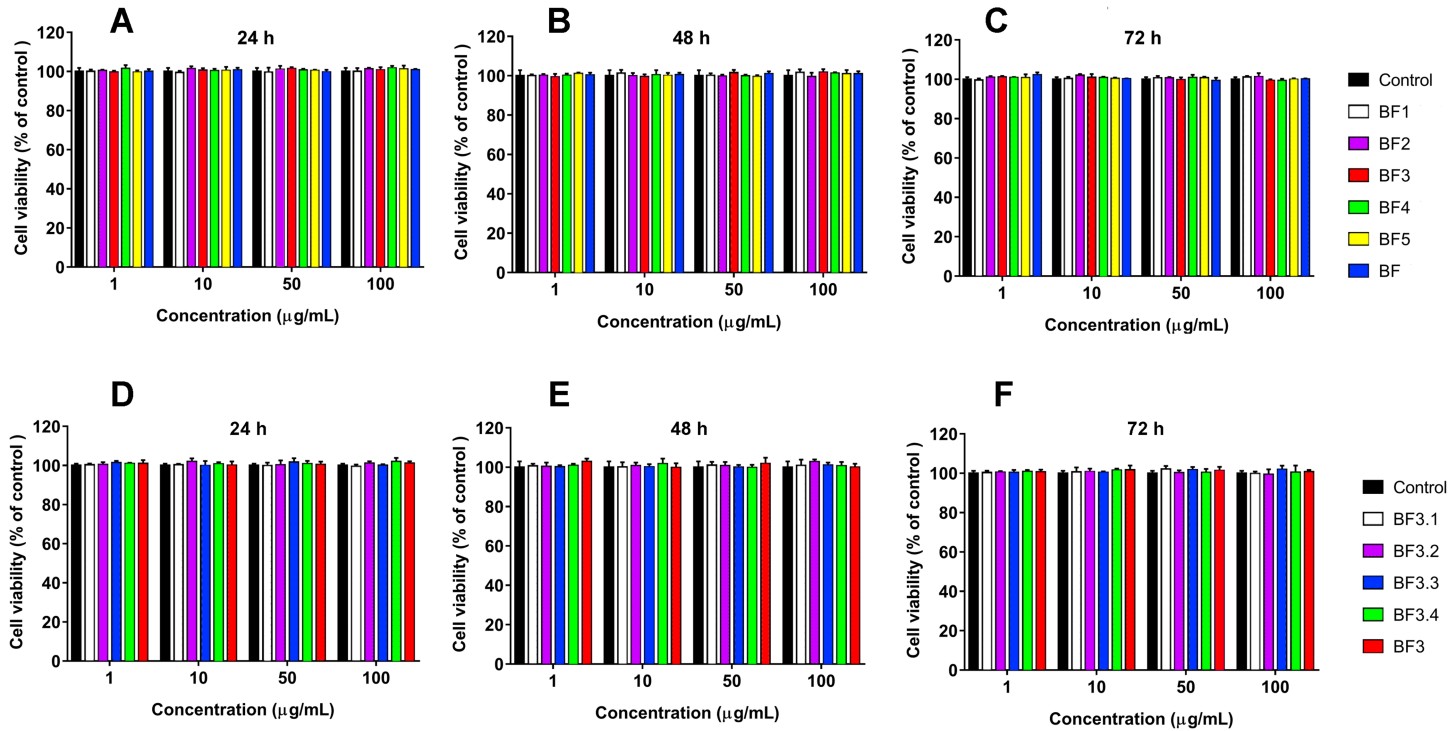

**Figure 4 Cytotoxicity test of fractions obtained from *B. fasciatus* venom against normal human kidney HK2 cells.** Cell viability was examined using the colorimetric MTT assay. The percentage of viable cells was determined from a comparison with untreated control. All results are presented as the mean ± SEM (standard error of the mean). (A–C) Gel-filtration fractions. BF1–BF5 correspond to fractions 1–5 from Fig. 1A. BF—*B. fasciatus* venom. (D–F) Fractions obtained after reversed phase chromatography (Fig. 1B). BF3.1–BF3.4 correspond to fractions 3.1–3.4 from Fig. 1B. BF3 fraction 3 from Fig. 1A. The significance of differences between experimental and control groups was analyzed by *t*-test: Two-Sample Assuming Equal Variances. No differences were found.

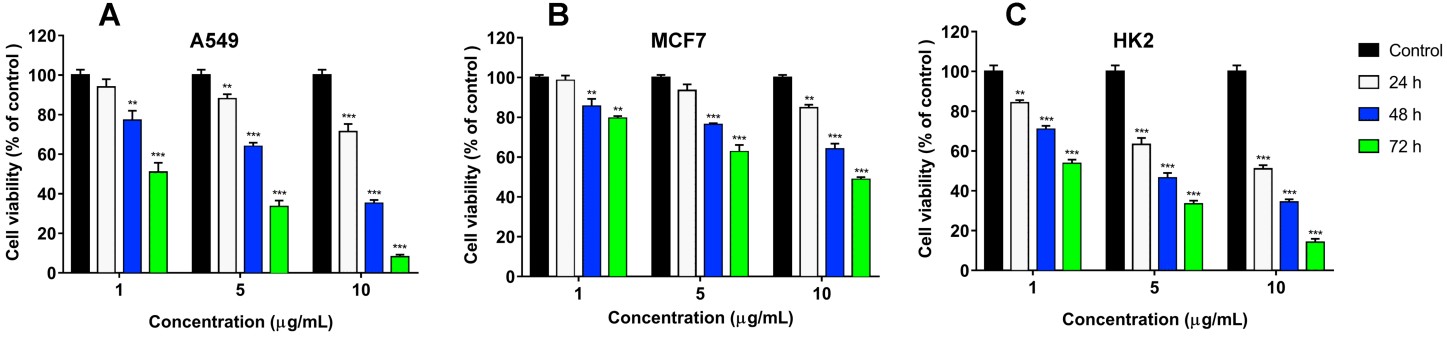

**Figure 5 Cytotoxicity of cisplatin to human A549 (A), MCF7 (B) and HK2 (C) cell lines.** Cell viability was examined using the colorimetric MTT assay. The percentage of viable cells was determined from a comparison with untreated control. All results are presented as the mean ± SEM (standard error of the mean). The significance of differences between experimental and control groups was analyzed by *t*-test: Two-Sample Assuming Equal Variances. Here $p < 0.01$ and $p < 0.001$ are indicated by ** and ***, respectively.

right side of the main intensive signal weak signals (like shoulders) corresponding to proteins with slightly higher molecular masses are seen. These signals represent ion adducts (+sodium or/and potassium ions) characteristic for the MALDI mass spectrometry or post-translationally modified proteins. For example, recently analyzing the proteome of the *Naja kaouthia* cobra venom we have found a number

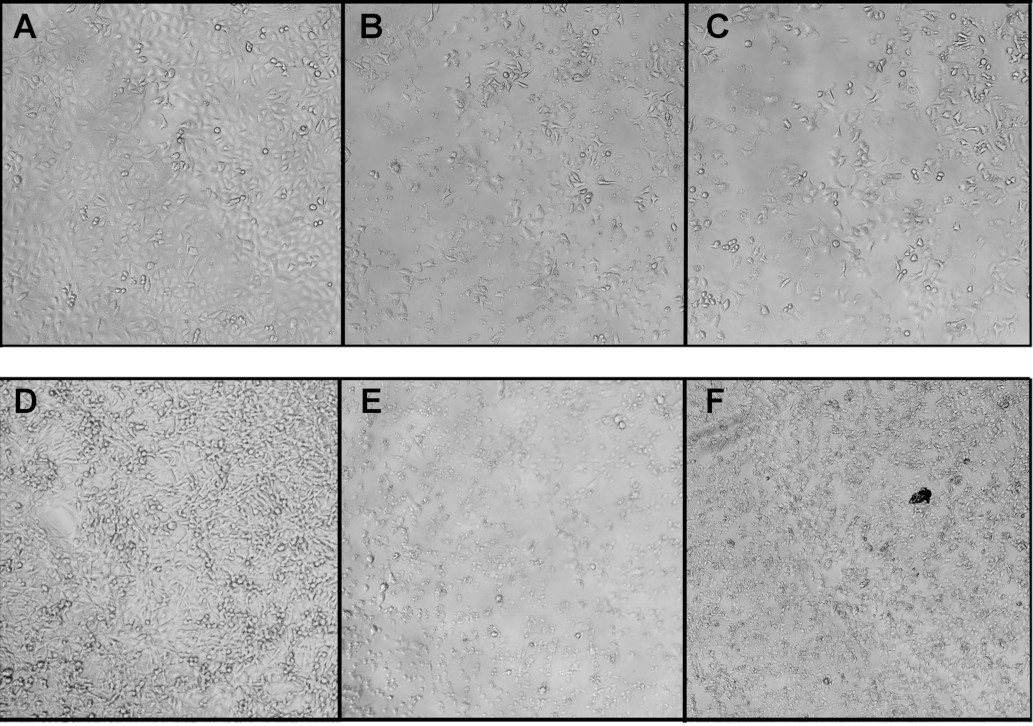

**Figure 6 Morphological changes of the A549 (A–B) and MCF7 (D–F) cells after treatment with fractions BF3 (B and E) or BF3.3 (C and F) for 72 h detected by phase contrast microscopy.** Changes in the cell morphology were observed using phase contrast microscopy with Zeiss Axio Vert 25C. Here are the representative images of cells untreated (A and D) or treated with fractions BF3 (B and E) or BF3.3 (C and F) at 100 µg/ml. All experiments were performed in triplicate and gave similar results.

of toxins, containing post-translational modifications (*Ryabinin et al., 2019*). After re-chromatography of fraction 3.3 by reversed phase HPLC using smoother gradient, the protein obtained was subjected to high-resolution mass-spectrometry analysis. Similarly to the results of MALDI mass spectrometry analysis, high resolution mass spectrometry revealed the main component accompanied by less intensive signals (Fig. 7A). Figure 7 shows fragments of the high-resolution mass spectrum of the protein sample; in this spectrum each protein molecule is represented by a pattern of peaks having a distribution close to Gaussian and representing a set of carbon isotopomers. The most intensive signal of izotopomer at m/z 1310.08897 (Fig. 7A) corresponds to molecular mass of 13090.89 Da, while the most abundant izotopomer of the more heavy protein at m/z 1311.68909 corresponds to the molecular mass of 13106.89 Da. The difference between these two proteins is equal to 16 Da, which is the result of the methionine oxidation. The signals at m/z 1313.38549 and 1314.88595 correspond to double and triple oxidized products. The signals with m/z values lower than the main component may correspond to the protein degradation products.

The isolated protein was digested with trypsin and the digest obtained was analyzed by LC-MS/MS. The analysis revealed that the digested sample is a basic phospholipase $A_2$ (Tables S1 and S2). As there are several very close homologs of phospholipase $A_2$ in
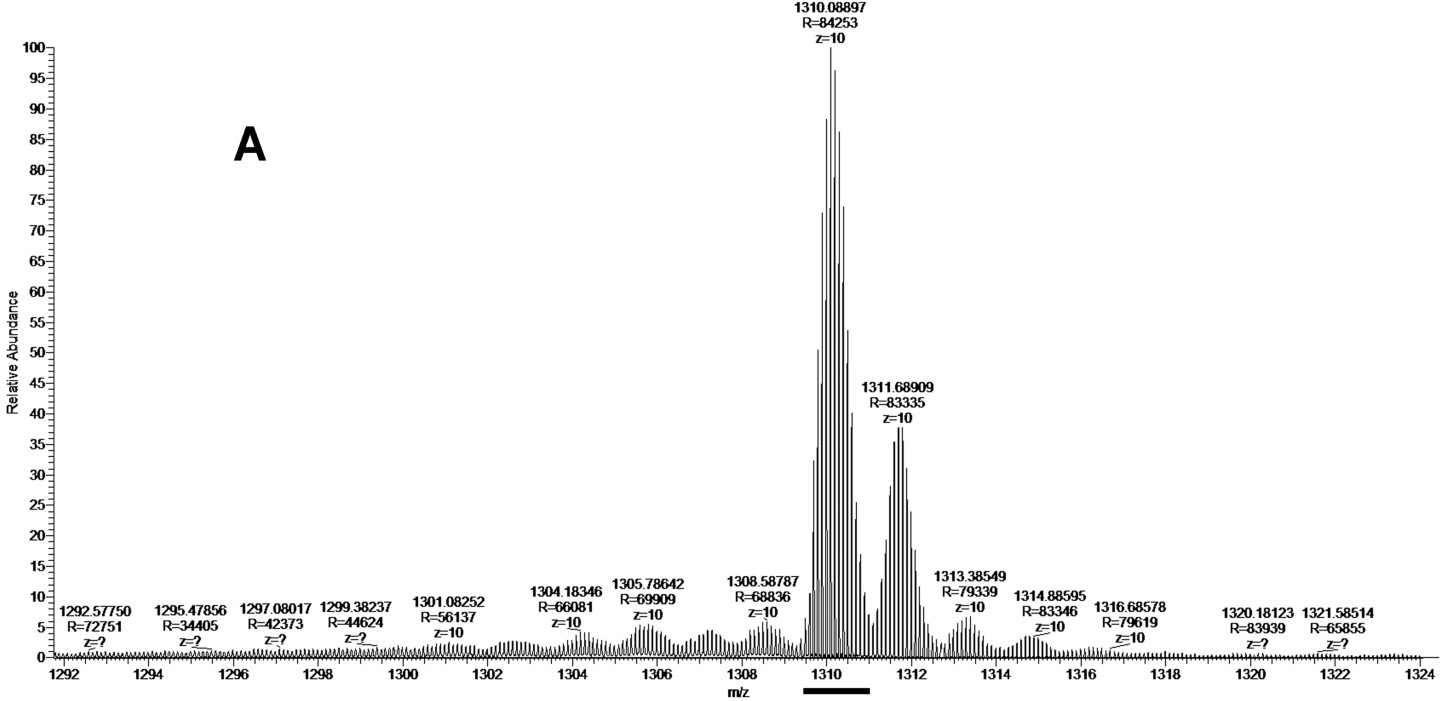

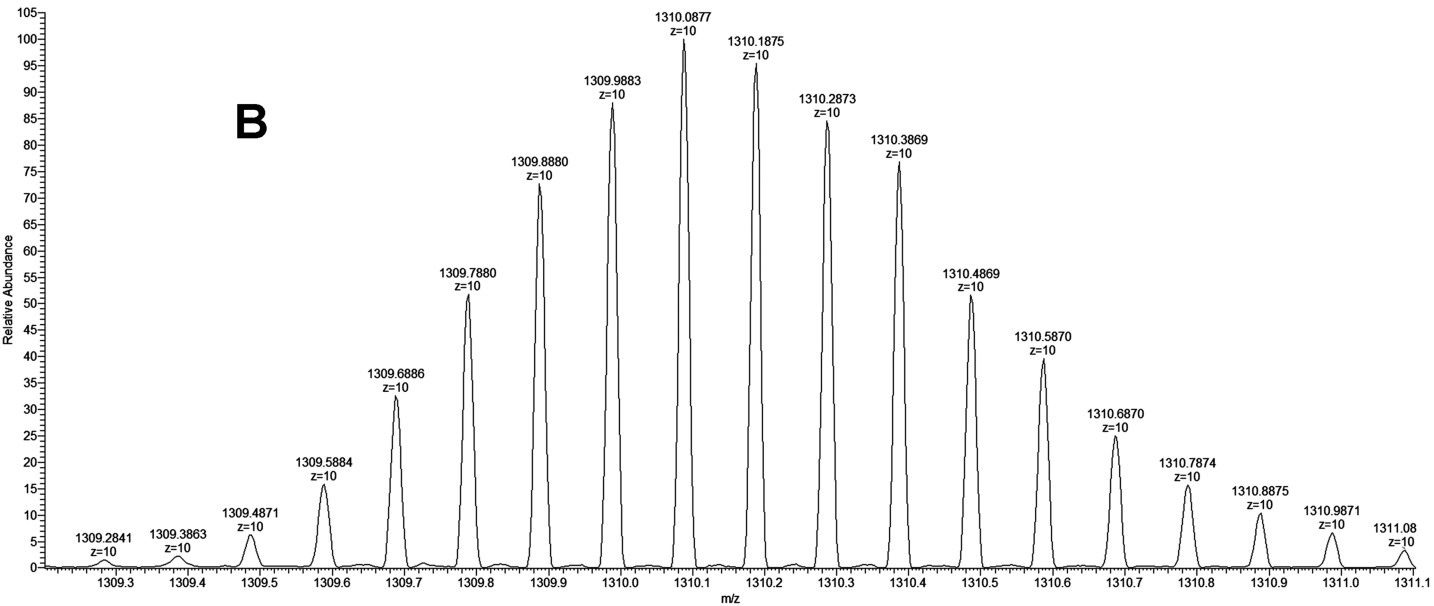

**Figure 7 The fragment of high-resolution mass spectrum of the protein from the fraction BF3.3.** (A) High-resolution mass spectrum at $Z = 10$. The horizontal bar indicates the isotopomers corresponding to the main component. (B) Enlarged fragment of the spectrum corresponding the main izotopomers ($Z = 10$).

*B. fasciatus*, the tryptic peptides were assigned to a few of them (Fig. S1). The most represented by the number of identified peptides protein was basic phospholipase A2 1 (UniProt KB accession number Q90WA7, 28–145 chain), 100% of its sequence was covered by identified peptide fragments (Fig. S1). This assignment was confirmed by

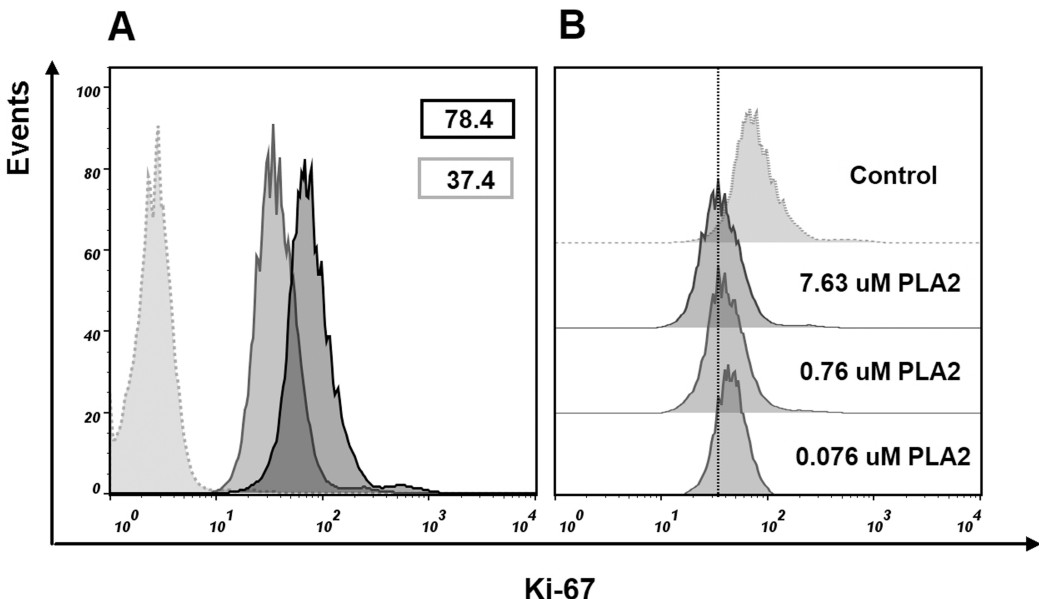

**Figure 8 Flow cytometry analysis of Ki-67 expression in MCF7 cells.** Cells were treated with of PLA$_2$ for 24 h then stained with Ki-67 antibody and analyzed by flow cytometry. Histograms show differences in the expression level of Ki-67 in phospholipase A$_2$ (PLA$_2$) treated and control cells. (A) Gray dotted histogram—isotype control; black histogram—control cells; gray histogram—cells treated with 7.63 μM of phospholipase A$_2$. The numbers in the histograms represent the mean fluorescence intensities in a population of Ki-67 positive cells. (B) Changes in the level of expression of Ki-67 in control cells and cells treated with different doses of phospholipase A$_2$. A dose-dependent decrease in the level of cell fluorescence with increasing concentration of phospholipase A$_2$ is shown.

monoisotopic mass determination. Assuming that the least intensive signal at m/z 1309.2841 (Fig. 7B) represented the monoisotopic mass of the protein under study, the corresponding mass was calculated to be qual to 13082.84 Da. This value with an accuracy of 5.35 ppm corresponded to theoretical value of 13082.91 Da for basic phospholipase A2 1 (Q90WA7). Basing on these data we suggested that protein manifesting the cytotoxicity to cancer cells was basic phospholipase A2 1 (Q90WA7).

## Flow cytometry studies

To find if isolated phospholipase A$_2$ affect the cell proliferation, we evaluated Ki-67 expression in MCF7 cells treated with this protein. After treatment with 7.63 μM of phospholipase A$_2$ for 24 h, flow cytometry analysis revealed a decrease in the proportion of Ki-67 positive cells (Fig. 8). As Ki-67 protein is a cellular marker for proliferation, its decline indicates the reduction in the proliferation of MCF7 cells treated with the phospholipase A$_2$. The noticeable changes in Ki-67 expression were also observed when the cells were treated with lower doses of the phospholipase A$_2$ (Fig. 8).

To examine whether cancer cells undergo apoptosis or necrosis after phospholipase A$_2$ treatment, the MCF7 cells were stained with fluorescently labeled annexin V and propidium iodide. Flow cytometry analysis of stained cells allowed to discriminate cells into four groups, namely viable (Annexin V$^-$/PI$^-$), early apoptosis (Annexin V$^+$/PI$^-$), late

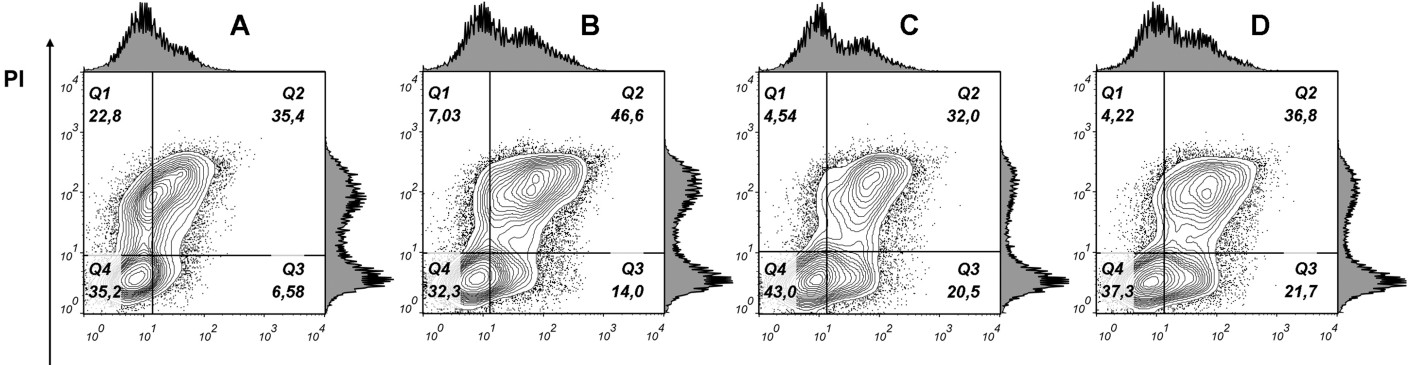

**Figure 9 Flow cytometry analysis of MCF7 cells treated with various doses of phospholipase A₂ for 24 h.** (A) Untreated cells. (B), (C), (D) Cells treated with 7.63, 0.76 and 0.076 µM of phospholipase A₂, respectively. Representative figures show population of viable (Q4, Annexin V⁻/PI⁻), early apoptotic (Q3, Annexin V⁺/PI⁻), late apoptotic (Q2, Annexin V⁺/PI⁺) and necrotic (Q1, Annexin V⁻/PI⁺) cells. Exposure to PLA₂ induces apoptosis in the MCF7 breast carcinoma cell line.

apoptosis (Annexin V⁺/PI⁺) and necrotic (Annexin V⁻/PI⁺) cells. In cells treated with phospholipase A₂ (7.63 µM for 24 h), we detected an increase in the Annexin V⁺/PI⁺ and Annexin V⁺/PI⁻ cell subpopulations (Fig. 9) indicating apoptotic cell death. After incubating cells with lower doses PLA₂, early apoptosis (Annexin⁺/PI⁻) rate tripled compared to untreated MCF7 cells, whereas late apoptosis (Annexin V⁺/PI⁺) subpopulation was changed a little. As phospholipase A₂ treatment resulted in increase in the proportion of apoptotic cells in MCF7 cells our studies show that apoptosis is a probable mechanism of cell death.

## DISCUSSION

In this paper, the isolation from *B. fasciatus* krait venom and characterization of phospholipase A₂ possessing cytotoxicity against the human breast cancer cells MCF7 and lung cancer cells A549 is described.

### Protein isolation and structural analysis

Kraits belong to the Elapidae family and most of the species from this genus produce neurotoxic venoms, the neurotoxic effects being caused basically by three finger toxins, which are one of the main components in these venoms. *B. fasciatus* is probably an exception as its venom contains relatively small amount of three finger toxins. In the *B. fasciatus* venom from Vietnam, the content of three finger toxins was about 1% and phospholipases A₂ was the main component representing about 70% of all proteins (*Ziganshin et al., 2015*). To isolate the active component from the venom, the activity guided purification scheme was used. The gel-filtration was used as a first step (Fig. 1A) and one predominant peak (fraction 3) was obtained. This fraction was cytotoxic to cancer cell lines and according to MALDI mass spectrometry contained proteins with molecular masses around 13 kDa (Fig. 3). We suggest that these are phospholipases A₂, as no other proteins with close molecular masses were found in this venom

(*Ziganshin et al., 2015*). The similar gel-filtration profile with one predominant fraction containing phospholipases $A_2$ was obtained during the study of *B. fasciatus* venom from India (*Tsai et al., 2007*). In our work, the cytotoxic fraction 3 was further separated by reversed-phase chromatography to produce three main fractions (Fig. 1B) which were analyzed by MALDI mass spectrometry (Fig. 3). All three fractions contained proteins with molecular masses around 13 kD which were similar to masses observed in fraction 3. As discussed in the Result section, in MALDI mass spectra in addition to main m/z signal some smaller signals were observed at higher molecular masses. We assigned these small signals to ion adducts characteristic for MALDI mass spectra and to post translationally modified toxins, in particular, to products with oxidized methionine. The methionine oxidation is a common protein modification and, for example, phospholipase $A_2$ with oxidized methionine was isolated from *Bothrops asper* snake venom (*Fernández et al., 2010*). Moreover, recently we have found that many toxins in *Naja kaouthia* cobra venom are subjected to post translational modifications. According to MALDI mass spectroscopy, fractions 3.2 and 3.3 contained a single protein each, while fraction 3.1 might contain several proteins. The cytotoxicity studies of HPLC fractions showed that only fraction 3.3 containing the protein with molecular mass of 13,093 Da was toxic to cancer cells. This fraction was analyzed further by high-resolution mass spectrometry. Structural analysis of isolated protein showed it was basic phospholipase A2 1 (Q90WA7). It should be mentioned that for *B. fasciatus* several phospholipases $A_2$ were identified; some enzymes were isolated from the venom, for others cDNA were cloned from the venom gland and sequenced. Comparison of the sequence established in this work with those in UniprotKB knowledge database showed its similarity to basic phospholipase A2 1. The amino acid sequence of this enzyme earlier was deduced from cloned cDNA and it is isolated from the *B. fasciatus* venom as a protein for the first time.

## Cytotoxic activity

The antitumor effect of snake venoms is well known. Their cytotoxity was demonstrated on several transformed cell lines, including HL-60 (human promyelocytic leukemia), HepG2 (human hepatoma), PC12 (adrenal pheochromocytoma), B16F10 (melanoma), Jurkat (acute T cell leukemia), SKBR-3 (human breast cancer), MCF7 (human breast cancer), A549 (human lung cancer) and some others. Several proteins possessing antiproliferative activity were isolated from snake venoms (*Li, Huang & Lin, 2018*). Thus, disintegrins, L-amino acid oxidases, metalloproteinases, and phospholipases $A_2$ were shown to possess the cytotoxicity to cancer cells. These toxins demonstrated different cytotoxicity depending on cell lines, being more toxic to certain cancer cells.

Human lung and breast cancers are among the deadliest cancer types, and there is a great need in development of new effective drug to combat these diseases. We have studied the effects of the whole *B. fasciatus* venom on the human breast cancer cells MCF7 and lung cancer cells A549 and observed cytotoxic effects against both cell lines (*Tran et al., 2019*). In this work targeted isolation of cytotoxic compound was carried out by means of liquid chromatography. As a result of gel-filtration followed by reversed phase chromatography, a protein was isolated which manifested the time- and

dose-dependent cytotoxicity for MCF7 and A549 cells while had no effect on HK2 normal cells (Fig. 4). Using high resolution mass spectrometry we showed that the amino acid sequence of the isolated protein coincided with that of phospholipase $A_2$ earlier identified in *B. fasciatus* at transcription level. After 72 h of incubation at concentration of 100 µg/mL (7.63 µM) it reduced the number of survived MCF7 and A549 cells by about 50% (Fig. 2). Therefore, this concentration can be considered as approximate IC50.

Morphological changes in the cells treated with different concentrations of crude venom of different fractions were observed using the inverted phase contrast microscope. After treatment with the venom or fractions, cells were detached from the plate and shrunk; the cell shape became rounded and intercellular spaces increased. These phenomena were observed for the isolated phospholipase $A_2$ (fraction BF3.3) as well (Fig. 6). Thus anticancer effect is clearly seen. While some morphological changes observed by microscopy, that is, cell shrinkage and membrane bubbling are typical for apoptosis, the molecular mechanism of phospholipase $A_2$ action was studied in more details by flow cytometry.

Marker protein Ki-67 is one of the most important indicators of the cell proliferation (*Pathmanathan & Balleine, 2013*). Our studies showed that the treatment of MCF7 cells with phospholipase $A_2$ resulted in decrease of Ki-67 expression in the treated cells. These data suggest anti-proliferative effect of phospholipase $A_2$ studied. Flow cytometry studies allows to discriminate precisely between apoptotic and necrotic cell death pathways. Using this method we observed that the phospholipase $A_2$ treatment of MCF7 cells resulted in the increase of apoptotic cells, thus suggesting the apoptotic death pathway.

It should be mentioned that a cytotoxic activity for several phospholipases $A_2$ from snake venoms was earlier reported (see, e.g., review *Sobrinho et al., 2016*). The activity observed strongly depended on the phospholipase $A_2$ nature and the cell line used for study (Table S3). However, the IC50 values are mostly in the range from about 40 to 200 µg/ml. The cytotoxicity of krait phospholipase $A_2$ found in this work is in the same range. The phospholipases listed in the Table S3 were isolated mostly from the venoms of viperid snakes. The data about cytotoxicity to cancer cells for phospholipases from elapid snakes are not so numerous. There are only two examples—nigexine from cobra *Naja nigricollis* (*Chwetzoff et al., 1989*) and phospholipase $A_2$ from see snake *Lapemis hardwickii* (*Liang et al., 2005*). Both these enzymes were somewhat more active than ours with IC50 values in the range from 39 to 69 µg/ml. However, these values were determined on the cell lines different from those used in our work.

Phospholipases $A_2$ are multifunctional toxins, interfering different biological processes. It was suggested earlier that to perform multiple task, they bind with high affinity to specific proteins which act as receptors, and a "pharmacological site" which is independent of the catalytic site should be presented on the surface of phospholipase $A_2$ molecule (*Kini, 2003*). In this work, we tested the effect of isolated protein on noncancerous cells and found that these cells were not affected by the isolated protein. This hardly could be possible if the cytotoxicity was the result of enzymatic activity. Moreover, it was shown earlier (*Bazaa et al., 2009*) that chemical modification with p-bromophenacyl bromide

abolished the enzymatic activity of phospholipase $A_2$ from *Macrovipera lebetina transmediterranea* without affecting its anti-tumor effect. These facts mean that cytotoxicity is not directly related to the enzymatic activity. The pharmacological site responsible for cytotoxic activity was localized to C-terminal region of the phospholipase $A_2$ molecule (*Araya & Lomonte, 2007*). Indeed, several synthetic peptides representing fragment 115–129 of phospholipase $A_2$ sequences manifested anticancer effects on various cancer cell lines (*Costa et al., 2008*; *Gebrim et al., 2009*; *Lomonte, Angulo & Moreno, 2010*). However, in general the cytotoxic activity of the synthetic peptides was lower than that of original phospholipases $A_2$. All the peptides represented fragments of phospholipases $A_2$ from snakes of Viperidae family. The amino acid sequences of C-terminal fragments of phospholipases $A_2$ from elapid snakes differ greatly from those of viperids. However, the cytotoxic activity of elapid phospholipases $A_2$ is very similar to that of viperid snakes. Moreover, C-terminal amino acid sequences are quite different between cobra and krait phospholipases $A_2$. This means that C-terminal fragment may not be so important for cytotoxicity in elapid phospholipases $A_2$.

It should be mentioned that the cytotoxicity against MCF7 and A549 cells were manifested by several snake venom proteins other than phospholipases $A_2$. Cytotoxins from cobra venoms (*Ebrahim et al., 2015*; *Attarde & Pandit, 2017*), ruviprase from Russell's viper (*Thakur et al., 2016*), and L-amino acid oxidases (*Li et al., 2014*; *Salama et al., 2018*) are found to inhibit the proliferation of MCF7 cells with different potency. The highest activity demonstrated L-amino acid oxidase from king cobra venom with an IC50 value of 0.04 µg/mL in MCF7 after 72 h treatment (*Li et al., 2014*). More diverse array of snake venom proteins manifested anticancer activity against A549 cells. These were serine proteases (*Nalbantsoy et al., 2017*), a low molecular weight C-type lectin daboialectin (*Pathan et al., 2017*), toxin C13S1C1 and toxin F-VIII of three finger toxin family (*Conlon et al., 2014*), as well as oxidases of L-amino acids (*Li et al., 2014*; *Wei et al., 2009*). Again, the most active was L-amino acid oxidase from king cobra venom with an IC50 value of 0.05 µg/mL in A549 after 72 h treatment (*Li et al., 2014*).

Considering the activity of snake venom proteins discussed above, one should take into account not only cytotoxicity to cancer cells, but their other biological effects and structural features. For example, L-amino acid oxidase is a very large protein and hardly can be used as drug, cytotoxins from cobra venom possess very high toxicity in vivo while disintegrins and C-type lectin-like proteins affect platelet functions. All these proteins may be regarded as a basis for the development of new medicines. Peptides derived from amino acid sequences of some phospholipases $A_2$ and possessing cytotoxicity to the cancer cells have been already discussed (*Costa et al., 2008*; *Gebrim et al., 2009*; *Lomonte, Angulo & Moreno, 2010*). Similar approach can be used for the other snake venom proteins including phospholipase $A_2$ from *B. fascistus* venom, described in this work.

## CONCLUSIONS

Basing on the earlier data about cytotoxicity of krait *B. fasciatus* venom to human cancer cells, we have isolated a basic phospholipase $A_2$ possessing cytotoxic activity. The results

obtained in this work demonstrated for the first time that a basic Asp49 phospholipase $A_2$ from *B. fasciatus* venom was able of exerting a cytotoxicity on human breast cancer MCF7 cells and human lung adenocarcinoma A549 cells but was not toxic to human kidney normal HK2 cells. Flow cytometry studies suggested that krait phospholipase $A_2$ had the anti-proliferative effect on MCF7 cells and induced apoptotic cell death pathway. The cytotoxic activity of krait phospholipase $A_2$ is close to that of other phospholipases $A_2$ isolated from viper and cobra venoms. However, the amino acid of krait toxin at the C-terminal region which was suggested to be important for cytotoxic effects differs strongly from that of other toxins. This means that apparently other sites of krait phospholipase $A_2$ are important for cytotoxicity to cancer cells. The identification of these sites deserves further study.

### Funding

The work was supported by the Vietnam Academy of Science and Technology (research project QTRU01.03/18-19) and Russian Foundation for Basic Research (Project No: 18-54-54006). The funders had no role in study design, data collection and analysis, decision to publish, or preparation of the manuscript.

### Grant Disclosures

The following grant information was disclosed by the authors:
Vietnam Academy of Science and Technology Research Project: QTRU01.03/18-19.
Russian Foundation for Basic Research Project No: 18-54-54006.

### Competing Interests

The authors declare that they have no competing interests.

### Author Contributions

- Thien V. Tran conceived and designed the experiments, performed the experiments, analyzed the data, prepared figures and/or tables, authored or reviewed drafts of the paper, approved the final draft.
- Andrei E. Siniavin conceived and designed the experiments, performed the experiments, analyzed the data, prepared figures and/or tables, authored or reviewed drafts of the paper, approved the final draft.
- Anh N. Hoang conceived and designed the experiments, analyzed the data, prepared figures and/or tables, authored or reviewed drafts of the paper, approved the final draft.
- My T.T. Le performed the experiments, authored or reviewed drafts of the paper, approved the final draft.
- Chuong D. Pham performed the experiments, analyzed the data, prepared figures and/or tables, approved the final draft.
- Trung V. Phung performed the experiments, analyzed the data, prepared figures and/or tables, approved the final draft.

- Khoa C. Nguyen conceived and designed the experiments, analyzed the data, authored or reviewed drafts of the paper, approved the final draft.
- Rustam H. Ziganshin performed the experiments, analyzed the data, prepared figures and/or tables, approved the final draft.
- Victor I. Tsetlin conceived and designed the experiments, authored or reviewed drafts of the paper, approved the final draft.
- Ching-Feng Weng conceived and designed the experiments, analyzed the data, authored or reviewed drafts of the paper, approved the final draft.
- Yuri N. Utkin conceived and designed the experiments, performed the experiments, prepared figures and/or tables, authored or reviewed drafts of the paper, approved the final draft.

## Data Availability

The raw measurements are available in the Supplemental Files.

## Supplemental Information

Supplemental information for this article can be found online at http://dx.doi.org/10.7717/peerj.8055#supplemental-information.

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
