# Peer review of "Phospholipase A2 from krait Bungarus fasciatus venom induces human cancer cell death in vitro"

_PeerJ, doi:10.7717/peerj.8055_

## Round 0.1 · original submission · Major Revisions

A number of issues have been pointed out by reviewers (attached). In particular, they coincide that cytotoxicity of the PLA2 should also be tested on a non-neoplastic cell line. This is an important weakness of the study. Also, some of the Tables and Figures are considered not relevant to the main body of this communication, and could be deleted or presented as supplementary material. In general legends to figures should be improved to be self-explanatory. Mass spectra can provide an excellent evaluation of protein homogeneity, but deconvolution should be presented (this may replace a gel electrophoresis image). I may add the following observations to those pointed out by reviewers:

1.Title: add "in vitro" at the end, to be more accurate and specific

2.Abstract: please refer to PRIMARY structure (not structure in general, as this would be misleading)

3.Text: The PLA2 activity of the isolated protein was not tested. In their Discussion, the authors expect the PLA2 to possess a molecular site responsible for the cytotoxic activity, but neglect to discuss that cytotoxicity may be, instead, the result of enzymatic activity (membrane phospholipid hydrolysis). It should be straightforward to assess the role of enzymatic activity in the cytotoxicity effect of this enzyme, by using the classical pBPB inhibitor.

4.Text: the authors use the terms "anti-proliferative activity" and "cytotoxicity" as synonyms, while these have clearly different mechanistic implications in cancer biology. Revise and use accordingly, with accuracy in each case referred.

5.Table 1 is not really pertinent to the focus of this manuscript, and in any case it is not comprehensive. Either delete or move to supplemental.

6.Tables 2 and 3 are unnecessary. Figs 1 and 3 could be combined into a nice plate (A + B). Fig. 3 needs the gradient to be drawn.

7.Figs 6 and 7 could also be combined. In Fig.6 the deconvolution to obtain the isotope-averaged molecular mass would be more informative. Choice of colors in Fig.7 is not optimal to clarity.

8.Figure 9 is irrelevant, since it does not add information to the conclusions of the paper - delete

Reviewer 1 ·

Basic reporting

In this paper, the authors evaluated the cytotoxic effects of a new Phospholipase A2 from krait B. fasciatus venom on two cancer cell lines. The work was well done and the paper is well written. Literature references, as soon as, introduction and results demontrate the relevance of the work.

Experimental design

The authors must inform the Ethical statement (scientific committee of the university)

Validity of the findings

Because the authors did not tried to go further in analyzing some possible mechanisms for Phospholipase A2 effects, the results are still preliminary. In addition, there are some concerns that must be addressed before considering publication. My main concern is if the cytotoxicity may be similar for non-tumor cells and it may be a limitation of the potential applications, as no control with no-tumor cells are showed. A justification and at least a comment about it should be included in the discussion, and expressed as a potential limitation in the final conclusions, and in the abstract.

Additional comments

Minor comments:
Results
1- The authors revealed a protein with molecular mass of about 13 kDa possessing cytotoxicity. From my point of view, the authors should demonstrate the protein purity by using electrophoresis or Mass Spectrometry and not the way they did considering they isolated a new PLA2.

2-Figure 1. Please, full information should be provided in the figure legend regarding the purification of PLA2. For example: Gel filtration of Bungarus fasciatus crude venom on Superdex HR75 column. Crude krait B. fasciatus venom (30mg) separated by gel-filtration on the Superdex HR75 column 86 (1x50 cm) equilibrated with the 0.1 M ammonium acetate buffer (pH 6.2). Flow rate 0.5 ml/min and fractions (x mL) was coleted. The eluate was monitored by spectrophotometry (OD=280nm).
All figures for legends need to be standardized with complete information. So, we invite the authors to add the precise information.
3-Figure 2. B. fasciatus venom. The correct is B. fasciatus venom in italic.
4-Figure 4. Legend: Cytotoxicity of fractions obtained after reversed phase chromatography against human MCF7 and A549 cell lines: “The statistical results are reported in Tables 2 and 3”. This information can be removed and replaced by: All results were analyzed by t-Test and are presented as the mean ± SEM (standard error of the mean).
5-Figure 5
Legend: Authors say informed morphological changes of the A549 (A-B) and MCF7 (D-F) cells after treatment with “various fractions” of crude venom but they compared only two fractions (BF3 and BF3.3). Please, correct the information.

Reviewer 2 ·

Basic reporting

The manuscript entitled-“Phospholipase A2 from krait Bungarus fasciatus venom induces human cancer cell death” by Tran et al. describes the characterization of anticancer activity of a RP-HPLC fraction from Bungarus fasciatus venom against MCF-7 and A549 cells. The anticancer activity was assayed by MTT-based cytotoxicity assay. Although it is an interesting study; however, the manuscript in the present format shows some preliminary and incomplete data. To my opinion few new experiments have to be performed for a more meaningful and conclusive outcome of this study. Several potent anticancer drugs are available in the market; therefore, the opportunity of discovering a new anticancer agent from snake venom should be highlighted in the Introduction section of the manuscript.

Experimental design

1. The purity and mass of the most active RP-HPLC fraction 3.3 were not determined and therefore, purity of preparation is under a question. Purity and molecular mass of the PLA2 enzyme must be determined by SDS-PAGE as well as MALDI-ToF-MS analyses. Further, line 217-218. The LC-MS/MS data suggests that the sample is a heterogeneous mixture of several isoforms of basic PLA2s which further raised question on the purity of preparation. This is a single PLA2 enzyme or mixture of several isoforms? Studies have shown that a minor contamination of some other toxins in PLA2 preparation can make the latter more cytotoxic and hemolytic. Therefore, purity of preparation must be established.
2. Multiple sequence alignment of tryptic fragment of B. fasciatus PLA2 enzyme with homologous proteins deposited in the NCBI database should be shown, may be as a supplementary Table.
3. It has been claimed that RP-HPLC fraction is a PLA2 enzyme. I suggest that the enzyme activity must be determined by biochemical assay. Further, purity of preparation in each step of fractionation, protein yield, and specific activity should be shown in a tabular form. The SDS-PAGE image of protein(s) from each fraction, starting from crude venom, gel filtration to RP-HPLC fractionations, should be shown.
4. Gel filtration chromatographic profile (Fig. 1). The separation of venom proteins was very poor and therefore, most of the proteins were eluted in a single sharp peak at around 60 min suggesting elution of 3FTxs of venom in this peak. Theoretically, PLA2 should elute before 3FTxs. However, elution of PLA2 with 3FTxs suggests complex formation of venom proteins. I suggest re-fractionation of venom to get a better separation and do the PLA2 assay of each fraction. Further, the gel-filtration column may be calibrated with protein molecular markers.
5. No positive control (anticancer agent) was used in this study. Literature survey shows that anticancer activity of chemical drugs are much higher than the PLA2 under study. Further, anticancer activity of this PLA2 is found to be much lower as compared to many snake venom proteins/peptides showing cytotoxicity against breast cancer cells. Therefore, it is difficult to envisage the anticancer potency of this PLA2 enzyme and its application as anticancer agent in future. Authors need to take a note of this point and discuss in their manuscript.
6. What is the cytotoxicity of this PLA2 enzyme against normal cells? This should be assessed to show that it does not have any harmful effect on normal cells. However, if this PLA2 also shows cytotoxicity against normal (non-cancerous) cells than its application as anticancer agent is doubtful.
7. Snake venom phospholipase A2 enzymes are known to exert toxicity. I would suggest to determine the in vivo acute toxicity of this PLA2 in a rodent model; otherwise, the therapeutic use of this protein as anticancer agent, as proposed by authors, will raise questions and doubts in the mind of readers.

Validity of the findings

1. The venom was obtained from a single snake or represents a pooled venom from several snakes? If it was a pooled venom please indicate the number of snakes. Who has taxonomically identified the snakes?
2. The concentrations of PLA2 should be shown in molar mass rather than in µg/ml.
3. Line 116. The results are mean of how many experiments (n)? Please indicate.
4. Tables 2 and 3 should be shown as supplementary Tables.
5. Figs. 1 and 3. The X-and Y-axes labeling are missing.
6. Figs. 2 and 6 should be shown as supplementary figures.

Additional comments

The manuscript in the present format is not suitable for publication. Author may consider my suggestions for further improvement of this work.

---

## Round 0.2 · accepted · Accept

Major points addressed by reviewers have adequately been addressed and clarified. The manuscript has been largely improved by text additions/modifications, as well as by reorganization of the figures and supplemental material as suggested.